# Temporal Changes in Litterfall and Nutrient Cycling from 2005–2015 in an Evergreen Broad-Leaved Forest in the Ailao Mountains, China

**DOI:** 10.3390/plants12061277

**Published:** 2023-03-10

**Authors:** Shiyu Dai, Ting Wei, Juan Tang, Zhixiong Xu, Hede Gong

**Affiliations:** 1School of Geography and Ecotourism, Southwest Forestry University, Kunming 650224, China; 2Ailaoshan Station for Subtropical Forest Ecosystem Studies, Chinese Academy of Sciences, Jingdong, Puer City 676209, China

**Keywords:** litterfall production, elemental composition, nutrient cycle, subtropical forest, Ailao Mountain

## Abstract

The study of litter can provide an important reference for understanding patterns of forest nutrient cycling and sustainable management. Here, we measured litterfall (leaves, branches, etc.) from a wet, evergreen, broad-leaved forest in Ailao Mountains of southwestern China on a monthly basis for 11 years (2005–2015). We measured the total biomass of litter fall as well as its components, and estimated the amount of C, N, P, K, S, Ca, and Mg in the amount of litterfall. We found that: The total litter of evergreen, broadleaved forest in Ailao Mountains from 2005 to 2015 was 7.70–9.46 t/ha, and the output of litterfall differed between years. This provides a safeguard for the soil fertility and biodiversity of the area. The total amount of litterfall and its components showed obvious seasonal variation, with most showing a bimodal pattern (peak from March to May and October to November). The majority of litterfall came from leaves, and the total amount as well as its components were correlated with meteorological factors (wind speed, temperate and precipitation) as well as extreme weather events. We found that among years, the nutrient concentration was sorted as C > Ca > N > K > Mg > S > P. The nutrient concentration in the fallen litter and the amount of nutrients returned showed a decreasing trend, but the decreasing rate was slowed through time. Nutrient cycling was influenced by meteorological factors, such as temperature, precipitation, and wind speed, but the nutrient utilization efficiency is high, the circulation capacity is strong, and the turnover time is short. Our results showed that although there was nutrient loss in this evergreen, broad-leaved forest, the presence of forest litterfall can effectively curb potential ecological problems in the area.

## 1. Introduction

In forests, organic matter from plants that is returned to the soil surface (e.g., fallen leaves, branches, floral and fruit parts) is generally referred to as litterfall [1,2]. The amount and quality of forest litterfall plays an important role in the development of soil and the cycling of nutrients. For example, growing plants absorb nutrients needed for their own growth from the soil, e.g., carbon, nitrogen, phosphorus, potassium, and other elements, and then return those elements back to the soil in the form of litterfall decomposition [3]. Among the different C fluxes of the forest ecosystem, canopy litterfall is the main aboveground organic C flux that reaches the soil, affecting C cycling as well as maintaining soil fertility globally [4]. Hence, litterfall acts as an important link between the aboveground production of trees and the soil organic C stock. At the same time, the litterfall production changes with climate, forest type, stand age, and season [5,6,7]. As a result, the quantity and quality of litterfall, as well as the environmental factors that influence them, regulate how these material cycles within ecosystems. However, there is variation in the turnover rates of different elements in a given ecosystem, as well as variation in turnover across regions in different climatic zones. Thus, it is useful to study the nature of forest litterfall dynamics through time in order to gain a deeper understanding of patterns of variation in litterfall among seasons and across years, and how this contributes to variation in forest nutrient cycling.

Plants periodically shed parts of their biomass as litterfall, which transfers C and nutrients from plants back into the soils and is a key biogeochemical process within forests [4,8]. However, litterfall in forests is variable throughout the year, with less litterfall during the growing season than the non-growing season, The main seasonal pattern of presentation is that: unimodal, bimodal, or irregular pattern [9]. In deciduous forests, litterfall happens during the non-growing season when low temperatures stimulate plant leaf synthesis of abscisic acid, resulting in a high levels of leaf fall, this has been confirmed in many studies [10,11,12]. In addition to variation across the season and across different forest types, litterfall is also variable among years as a result of variation in climatic conditions (e.g., wind and snow) and forest age. As global climate change continues, studies on the within and among year variation in litterfall, and its role in nutrient cycling, will provide important baseline knowledge for understanding how these will change in the future [13,14,15,16].

Subtropical forests have high primary productivity and are also hotspots for biodiversity research., which play an important role in carbon storage in global terrestrial ecosystems [17]. The montane, moist, evergreen, broad-leaved forest in the Ailao Mountain Nature Reserve in Yunnan is currently the largest and most well-preserved subtropical evergreen broad-leaved forest in China. It is one of the valuable zonal vegetation. This is particularly urgent for understanding patterns of litterfall and nutrient cycling in primary forests which are rapidly disappearing. Hence, we can also better understand and utilize natural resources, thus improving the stability and sustainability of the ecosystem.

Here, we measured the total amount of litterfall, as well as its nutrient concentration, from monthly samples collected over an 11-year period (2005–2015) in the primeval forest of Ailao Mountain National Nature Reserve in Yunnan Province, southwestern China. We compared this variation at monthly, seasonal, and annual periods and examined how they were correlated with precipitation, temperature, wind speed, and extreme weather. Results from our study show considerable variation within and among years in the quantity and quality of litterfall in this forest, providing baseline data for studying forest nutrient cycles.

## 2. Results

### 2.1. Dynamic Characteristics of Litterfall and Its Component Output

#### 2.1.1. Interannual Dynamics of Litterfall and Its Component Output

Between 2005 to 2015, we found that the amount of litterfall per year ranged from 7.70 to 9.46 t/ha a (Figure 1, Table 1), with an annual average of 8.11 ± 0.73 t/ha a. Across the observation period, we found that deciduous leaf litter was the greatest component of the overall leaf litter (representing 42–62% of the litterfall). Other components of litterfall, including fruit/flower drop, bark, moss/lichen, and other debris, were observed to an intermediate (21–26% of the litterfall) extent, and litterfall from branches the least (17–32% of the litterfall) (Table 1).

#### 2.1.2. Monthly Variation in the Output of Litterfall and Its Components within a Year

There was clear seasonal variation in the total litterfall in this forest (Figure 2a), but the variation was not always consistent among years. For example, most years had multiple peaks of litterfall, with one peak around April (ranging from March–May) and another later in the October–November. There was variation in these peaks, however. One year, 2015, was unique with the highest observed litterfall of the whole time series occurring in January. When we analyzed litterfall in its components, we found that leaf litter (Figure 2b), as the most abundant component, largely mirrored that of the total litterfall biomass. The other two components of litterfall, branches (Figure 2c), and ‘other’ (Figure 2d) were more variable throughout the year. The main source of litterfall production is leaves and branches, with the highest production of branches in January and February, and the highest production of leaves in other months (Table 2).

#### 2.1.3. Correlation Analysis of Litterfall and Its Components with Climatic Factors

Overall, we found that the total litterfall was negatively correlated with the average wind speed, but positively correlated with temperature and precipitation (Table 3). These trends were mirrored by the ‘other’ category of litterfall, while only leaf litterfall was positively correlated with average monthly temperature (Table 3). For monthly data, we found that the total amount of litterfall and meteorological factors was positively correlated with monthly precipitation in the first 1–2 months, but there were no other positive correlations between litterfall production and monthly climatic variation (Table 4).

### 2.2. Dynamic Characteristics of Litterfall Nutrient Concentration

#### 2.2.1. Interannual Variation Characteristics of Litterfall Nutrient Concentration

We found high variation in the nutrient concentration of litterfall among years and in different components of the litterfall (Table 5). In general, the concentration of C, Ca, and Mg decreased from 2005 to 2010 and 2015, while the concentration of N, P, K, and S increased across this same period. While the nutrient concentration of litterfall was C > Ca > N > K > Mg > S > P in 2005 and 2010, the abundance of N increased relative to the other elements, such that C > N > Ca > K > Mg > S > P in 2015. Across years, we found higher nutrient concentration in leaves compared to branches.

#### 2.2.2. Characteristics of Intra-Year Variation of Litterfall Nutrient Concentration

We illustrate the within year variation of each element in the litterfall in Figure 3, showing that each element has a signature variation. C concentration is higher in the first half of the year and declines to its lowest level in late autumn and early winter (Figure 3a). N, P, and S showed similar variation across the year, first declining through the first months of the year, peaking in the middle of the growing season, and declining again towards fall and winter (Figure 3b–d). The last three elements showed less distinct patterns and fluctuated around mean values throughout the year (Figure 3e–g).

### 2.3. Dynamic Characteristics of Litterfall Nutrient Element Return

#### 2.3.1. Interannual Return Characteristics of Litterfall Nutrient Elements

We found that the return of leaf litter (except Ca) was higher than that of branches litter, and the return of C was much higher than that of other nutrient elements. In all, the returns were roughly sorted as C > N > Ca > K > Mg > S > P (Table 6). Overall, we found a downward trend of the annual mean return of nutrients from 2005 to 2015. In branches, the return of C, N, and S decreased through time, while the return of P and K increased; the return of Ca and Mg first decreased and then increased.

#### 2.3.2. Characteristics of Intra-Year Return of Litterfall Nutrient Elements

In Figure 4, we show that the return of each element varies greatly throughout the year. The return of C is the largest, with a peak May. The return of N and S showed a multimodal distribution at the beginning of the year, mid-year, and at the end of the year, but there was a sudden decrease in July. K and P showed a peak in August. The return of Ca and Mg first decreased until around July and then increased.

In Table 7, we show that some, but not all, element returns are affected by meteorological factors. The returns of C, N, and S were not correlated with meteorological factors (temperature, precipitation, wind speed). The return of P and K was significantly negatively correlated with wind speed. The return of K was also positively correlated with precipitation, while the return of Ca was negatively correlated with precipitation. Finally, the return of Mg was negatively correlated with temperature and precipitation.

### 2.4. Biological Cycle of Nutrient Elements in Evergreen Broad-Leaved Forests of Mount Ailao

Nutrient cycling refers to the absorption of nutrients from the soil by plant, some of which is used for plant growth, while the rest is returned to the soil through litterfall, secretions and rainwater. This is given by absorption, retention, and return of the three links, where the cycle balance formula is: absorption = retention + return [18,19]. In our study, we were only able calculate the return of litterfall, thus underestimating the total cycle. Nevertheless, we can use a utilization coefficient, circulation coefficient, and turnover time to estimate elements of the cycle [20]. The nutrient utilization coefficients were distributed from 0.23 to 0.29, with an average value of 0.25, which showed Ca > Mg > S, C > N, K > P. The circulation coefficients were distributed from 0.42 to 0.84, with an average value was 0.53, showing P > K > N > C > S > Mg > Ca. The turnover time was distributed from 8.40a to 14.14a, with an average value of 10.50a, which was manifest as Ca > Mg > S > C > N > K > P (Table 8). 

## 3. Materials and Methods

### 3.1. Overview of the Study Area

Our study area was located in the Xujiaba area (24°32′ N, 102°01′ E) within the Ailaoshan National Nature Reserve, Jingdong Yi Autonomous County, Puer City, Yunnan Province. The study area occurred at an altitude of 2400–2600 m, and the soil was fertile, acidic, yellow-brown soil [21]. The climate at the Ailao Mountain Forest Ecosystem Research Station is southwestern monsoon, with distinct dry and wet seasons (annual average temperature is 11.3 °C and the annual average precipitation is 1931 mm.), at the transition from central subtropical to south Asian tropics. Our study site is within the largest area of the primitive Zhongshan wet, evergreen, broad-leaved forest preserved in China, with a closed canopy and layered shrub layer with abundant epiphytes on the trees. The dominant tree species at the site include *Machilus bombycina*, *Populus rotundifolia*, *Schima noronhae*, *Castanopsis rufescens*, and *Castanopsis delavayi* [22].

### 3.2. Litterfall Sampling and Collection

We established a litterfall collection grid within a 1-ha long term observation plot of the forest. Specifically, we divided the plot into 100 10 m × 10 m subplots. From those 100 subplots, we randomly selected 25 and placed a 1 m^2^ litterfall collection basket in each. We constructed litterfall collection baskets out of steel frame boxes 1 m × 1 m × 0.25 m covered with 0.5 mm nylon mesh. We inserted the four corners of each basket into the soil such that the bottom of the basket was about 0.5 m from the ground.

We collected litterfall from each basket at the end of each month from 2005 to 2015. We sorted litterfall into categories, including branches, leaves, fallen flowers and fruits, bark, moss and lichen, and other debris, drying each in an oven at 65 °C to a constant weight, and then recorded the dry weight of each component.

### 3.3. Meteorological Data Observation

Meteorological data, including precipitation, temperature, and wind speed, were collected from the Ailao Mountain Meteorological Station. Data were averaged monthly and according to season (Figure 5). The wet season is from June to October, while the dry season is from November to May.

### 3.4. Measuring Nutrients in the Litterfall

We estimated important elements in the litterfall after drying by first grounding the littler into a fine powder, which was subsequently sieved through a 250-µm mesh. We determined carbon (C) and nitrogen (N) using a carbon analyzer (EA3000 EuroVector, Milan, Italy) [23,24]. To prepare samples for phosphorus (P) and potassium (K), we first digested samples in H_2_O_2_-H_2_SO_4_. We determined phosphorus (P) concentration using molybdenum antimony colorimetry, potassium (K) was measured using plasma atomic emission spectrophotometer [2], and the concentration of sulfur (S), calcium (Ca) and magnesium (Mg) was measured using a flame photometer and spectrophotometer [25]. 

### 3.5. Statistical Analysis of Data

We averaged the total litterfall output across the 25 collection baskets and took monthly and annual (sum of the 12 months) data for analyses. After using the Shapiro–Wilk test to test the normality of the data, one-way ANOVA and LSD were used to compare the difference in the amount of litterfall in different parts of different years and its components, the concentration of nutrient elements, and the amount of return. The ANOVA is mainly used to test whether there is a significant difference in the mean of the components of litterfall between the interannual and intra-annual periods. The use of multiple comparisons can support a better understanding of the differences between them. In addition, we also use SPSS26.0 to linear fit the environmental variables (temperature, precipitation, wind speed) and the output of components of litterfall and nutrient concentration, and then explore the correlation between them.

We calculated the annual return of litterfall nutrients as follows:(1)$$La=\sum_{i=1}^{12}\sum_{j=1}^{25}LijCij/100$$
where *L_a_* is the annual amount of nutrients returned by litterfall; *L_ij_* is the litterfall amount of the *j*th component in the *i*th month (kg/m^2^); *C_ij_* is the nutrient concentration (g/kg) of the *j*th component of litterfall in the *i*th month [26].

The biocirculating coefficient mainly includes nutrient utilization coefficient, cycle coefficient, and turnover time. The nutrient utilization coefficient is the ratio of the elements absorbed by the plant per unit time and unit area to the existing elements of the plant, and the calculation method of nutrient utilization is mainly based on the Chapin index.
*E =* A_p_/M(2)

It can be seen from Equation (2) that *E* is Chapin index, M is plant biomass, and A_p_ is nutrient storage (t/ha). In essence, Chapin index is the average content of plant body nutrients, which reflects the amount of nutrients consumed by the plant construction unit per unit biomass. However, Chapin index has a bias in overestimating the nutrient utilization efficiency of trees, and formula (2), revised from the perspective of nutrient cycle to obtain formula (3), can also better reflect the nutrient utilization status of trees.
*R*_e_ = F_a_/A_p_(3)
where *R*_e_ is the utilization coefficient, F_a_ is the nutrient uptake (t/ha a), and A_p_ is the nutrient storage (t/ha).

The nutrient cycle coefficient is a kind of index proposed based on the concept of the biological cycle, also known as the biological return coefficient. The method has certain limitations in reflecting the overall situation of forest nutrient cycling, since the calculation of the nutrient cycling coefficient does not involve the decomposition of forest litterfall, and the decomposition of litterfall is an important link in the nutrient cycle. The calculation is performed as follows:*R_g_ =* F_d_/F_a_(4)
where *R_g_* is the cycle coefficient, F_a_ is the nutrient uptake (t/ha a), and F_d_ is the nutrient return (t/ha a).

Turnover time is the time it takes for a nutrient element to go through one cycle.
*T_t_ =* F_d_/A_p_(5)
where *T_t_* is the turnover time, A_p_ is the nutrient storage (t/ha), and F_d_ is the nutrient return (t/ha a) [27].

## 4. Discussion

Litterfall volume is a component of the forest ecosystem biomass, which reflects the primary productivity level as well as the functions of the forest ecosystem [25,28]. The research shows that average annual litterfall of evergreen broad-leaved forest is 6.96 t/ha [29], the average annual litterfall of Yuanjiang savanna ecosystem is 2.5–3 t/ha [30], and the average litterfall of Chinese grassland is 0.59 t/ha [31]. It can be seen that fallen forest materials play a very important role in the global ecosystem. Our results showed that the average annual litterfall from 2005 to 2015 was 8.11 t/ha. This is similar to that observed in evergreen broadleaved forests in other subtropical regions (e.g., Dinghushan South Asian tropical evergreen broadleaved forest (7–11 t/ha) [32], and Xiaokeng subtropical evergreen broadleaf forest (7.99–8.45 t/ha) [33]. Furthermore, Guan Xin summarized the research results of central, subtropical, evergreen broadleaved forests and found that the annual litterfall recovery ranged from 3.90 to 7.72 t/ha [34]. Thus, the amount of litterfall observed at our site was intermediate between the South Asian tropical monsoon, evergreen, broad-leaved forest and the central, subtropical, evergreen, broad-leaved forest; its litterfall production is much higher than other ecosystems.

When compared to evergreen, broadleaf forests in other subtropical regions, we found that the ratio of leaf litter to total litter was less than that of Wuyishan rice oak forest (77.03 ± 1.93%) [35], Guangxi Longgang National Nature Reserve (85%) [36], Baishan Zu evergreen broadleaf forest (51.34%) [37] and Zhejiang Ningbo Tiantong Mountain evergreen broadleaved forest (50.7%) [38]. The high variability we observed within and among years for total litter and its components was consistent with the results of Zou Bingzhang [39]. Likewise, we found that most litterfall came from leaves with less from other sources, which was consistent with the results of Wan Chunhong [40]. The size and dynamic changes of forest litterfall output are influenced by many factors and are the result of a combination of factors [41,42]. We found that the overall amount of litterfall on the ground is high, which represents positive feedback for the forest ecosystem. It provides abundant food sources for forest organisms, especially fruits and flowers, which play an important role in the survival and reproduction of rodents. This also indirectly ensures the survival and reproduction of birds that feed on rodents. On the other hand, the litterfall decomposes and releases nutrients, keeping the fertility of the research site at a high level, as well as changing the physical properties of the forest soil. Litterfall also has a strong water retention capacity, which can reduce water evaporation and maintain sufficient water storage on the forest surface, which is important for water conservation and maintaining soil environment stability [43,44]. The presence of litterfall plays a foundational role in the entire forest ecosystem. We also found that litterfall dynamics correlated with a number of features of the environment, including wind speed, precipitation, and temperature. In the one exceptionally unusual month year in our survey (January 2015), branch content was high as a result of an unusual amount of snow in this period (1129.2 mm), which created considerable tree and branch fall in this period. This indicates that, after the interference of extreme ice and snow weather, the leaves were violently shaken by strong external forces, resulting in non physiological shedding.

The nutrient concentration of litterfall is related to the characteristics of the plants and the soil nutrient content [45]. We found that the nutrient concentration of litterfall was roughly the same in different years and different litterfall components. Specifically, we found that C concentration was the highest, Ca and N concentration were second, and K, Mg, S, and P concentrations were relatively low, consistent with the results of Chen Jinlei and Xue Fei [46,47]. Our finding that different nutrients varied through time is similar to the research results of Xue Fei and Zhao Chang [45,47]. Our results show that many elements are correlated with temperature and precipitation, which may be influenced by the relationship between these elements and plant growth [48]. For example, K is highly mobile, and P is easily affected by multiple factors, such as vegetative growth rhythms, rainfall leaching, microbial degradation, etc.

The return of litterfall nutrients to the soil is influenced by a combination of factors. We found that the return of nutrient elements in litterfall showed a decreasing trend through the year. Our annual nutrient return size of litterfall is roughly C > N > Ca > K > Mg > S > P, which differs from the results of Liu Yi, but is similar to those of Liu Lei and Gao Shilei [49,50,51]. Furthermore, the annual return amount of litterfall C in our study area (6.21 t/ha) was within the range of litterfall C return observed (0.05~7.50 t/ha) across the world’s forested ecosystems [52]. Likewise, our results regarding the return of C, N, and P were higher than those of Mijiao natural forest in Sanming City, Fujian Province and Wuyi Mountain evergreen, broadleaved forest [53,54]. This suggests that evergreen, broadleaf forest literfall in at our study site in the Ailao Mountain plays a very important role in the carbon cycle of the soil. Additionally, the litterfall results in modifications in forest ecosystems, particularly in subtropical, evergreen, broad-leaved forests because the amount of litterfall can regulate the micro-climate in the soil, affecting the decomposition rate with the changes in microbial community, and soil microorganisms in the soil affect soil respiration [55].

The utilization coefficient, circulation coefficient, and turnover time are all important parameters in the nutrient cycling process, and the nutrient cycling parameters also vary due to the difference between nutrient uptake and return by different forest types [56]. The utilization coefficient is the ratio of absorption to storage, reflecting the storage rate of ecosystem elements; the larger the coefficient, the greater the storage capacity of plants and the lower the utilization efficiency. The nutrient utilization coefficient in our study area was 0.25, which was lower than that found in the karst peak cluster depression in Huanjiang, Guangxi (0.35) and the four-year-old Mazhan Acacia plantation (0.51) in the state-owned peak forest farm in Nanning City, Guangxi [20,57]. This indicates that the forest in our study system had high nutrient utilization efficiency and low storage capacity. The circulation coefficient is the ratio of plant return to absorption, and reflects the size of the residual amount of the element during the cycle; the larger the coefficient, the faster the rate of element circulation and the greater the fluidity. The cycling coefficient in our study (0.53) was higher than that of Gongga Mountain Natural Forest (0.474) [58], but lower than that of Dinghu Mountain Horsetail Pine Forest (0.68) and *Pinus tabulosis* forest (0.71–0.85) in the loess hilly area [19,59]. This indicates that the forest in the study area had high nutrient cycling capacity. Turnaround time is the ratio of a plant’s total nutrient storage to return, indicating the time it takes for a nutrient element to go through the cycle. The longer the turnaround time, the longer the nutrients stay in plants. In this study, we found that the turnover time ranged from 8.40 a to 14.14 a (average value of 10.50 a), which was manifested as Ca > Mg > S > C > N > K > P, suggesting that Ca and Mg were inactive elements and P was the active element. Meantime, this also indicates that the plants in the research area absorb nutrients quickly, grow quickly, and have high yield and large biomass.

## 5. Conclusions

In this paper, we present the results of an 11-year study investigating the dynamic changes of litterfall output, nutrient concentration, and return in a wet, evergreen, broadleaf forest in the Ailao Mountains of China to draw the following conclusions:

From 2005 to 2015, the total litter of evergreen, broadleaved forests in the Ailao Mountains was 7.70–9.46 t/ha a, and the interannual fluctuation of litterfall was large, with an average of 8.11 ± 0.73 t/ha, which was higher than that of Central Asian, thermal, evergreen, broadleaved forests. The presence of a large amount of litterfall provides nutrients to the study area, promoting the development and stability of the study area’s ecosystem and ensuring the fertility of the soil and biodiversity. The output of litterfall was significantly different between different years. There are significant interannual and seasonal variations in the amount of litterfall, mostly bimodal (peak from March to May, October to November), with higher levels of leaf litter than other components in each year, some of which was correlated with meteorological factors (*p* < 0.05). The article only discussed the meteorological factor as the cause of the litterfall, indicating that the production of litterfall is the result of multiple factors. We generally found the nutrient concentration was sorted as C > Ca > N > K > Mg > S > P, except for the slight difference in Ca and N, this also conforms to the general pattern of changes in forest nutrient concentrations. The nutrient concentration and returned amount of litterfall are related to the growth of trees, as well as variation in meteorological factors, such as temperature, precipitation, and wind speed. Our results showed that although there was nutrient loss in the evergreen, broad-leaved forest area of the Ailao Mountains, forest litterfall could still maintain soil fertility in the area, maintaining the normal operation of the entire forest ecosystem.

## Figures and Tables

**Figure 1 plants-12-01277-f001:**
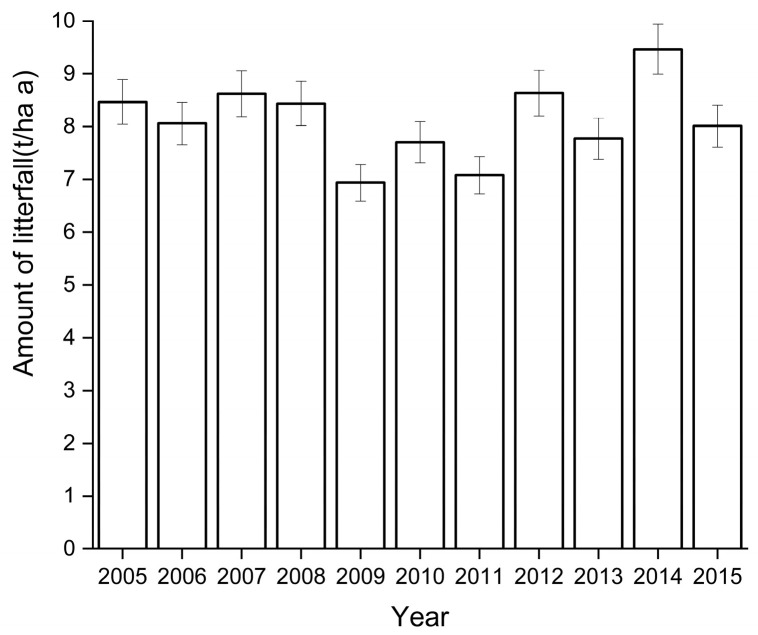
Interannual dynamics of litterfall from 2005 to 2015.

**Figure 2 plants-12-01277-f002:**
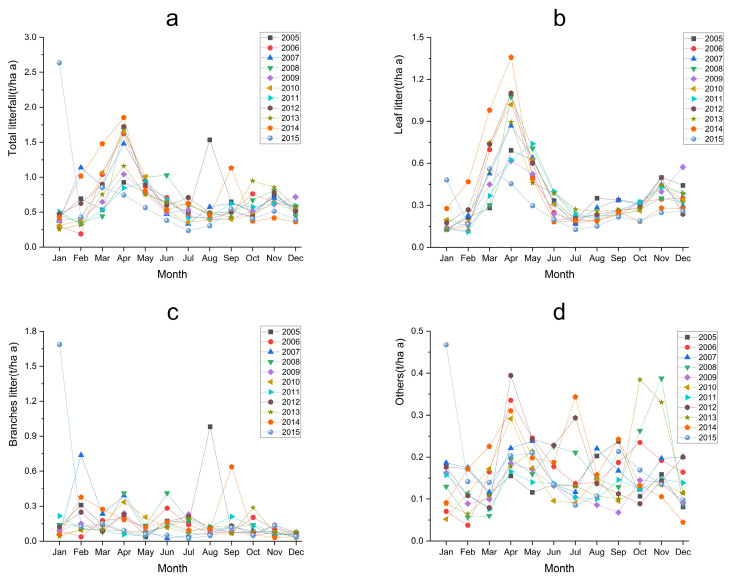
Monthly variation of litterfall and its component output in evergreen broadleaf forest in Ailao Mountain. (**a**) Total litterfall, (**b**) Leaf litter, (**c**) Branches litter, (**d**) Others.

**Figure 3 plants-12-01277-f003:**
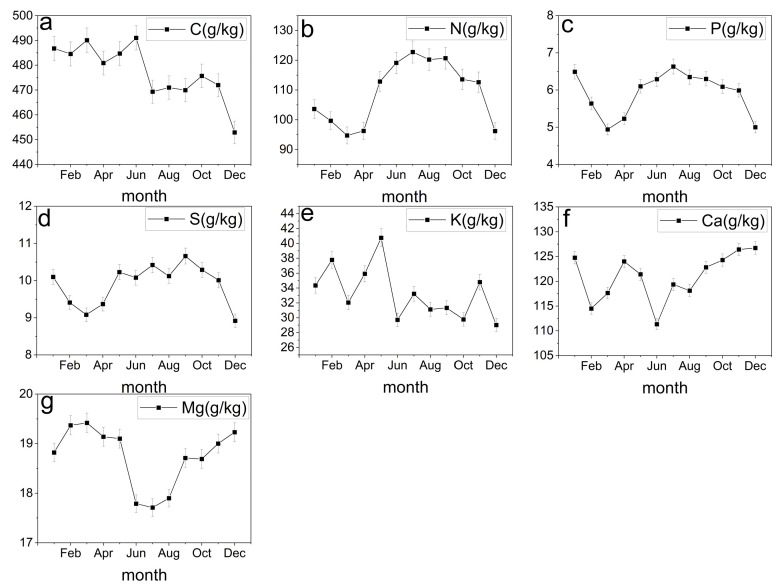
Within-year change of C, N, P, K, S, Ca and Mg concentrations in litterfall. (**a**) Within-year change of C element concentration. (**b**) Within-year change of N element concentration. (**c**) Within-year change of P element concentration. (**d**) Within-year change of S element concentration. (**e**) Within-year change of K element concentration. (**f**) Within-year change of Ca element concentration. (**g**) Within-year change of Mg element concentration.

**Figure 4 plants-12-01277-f004:**
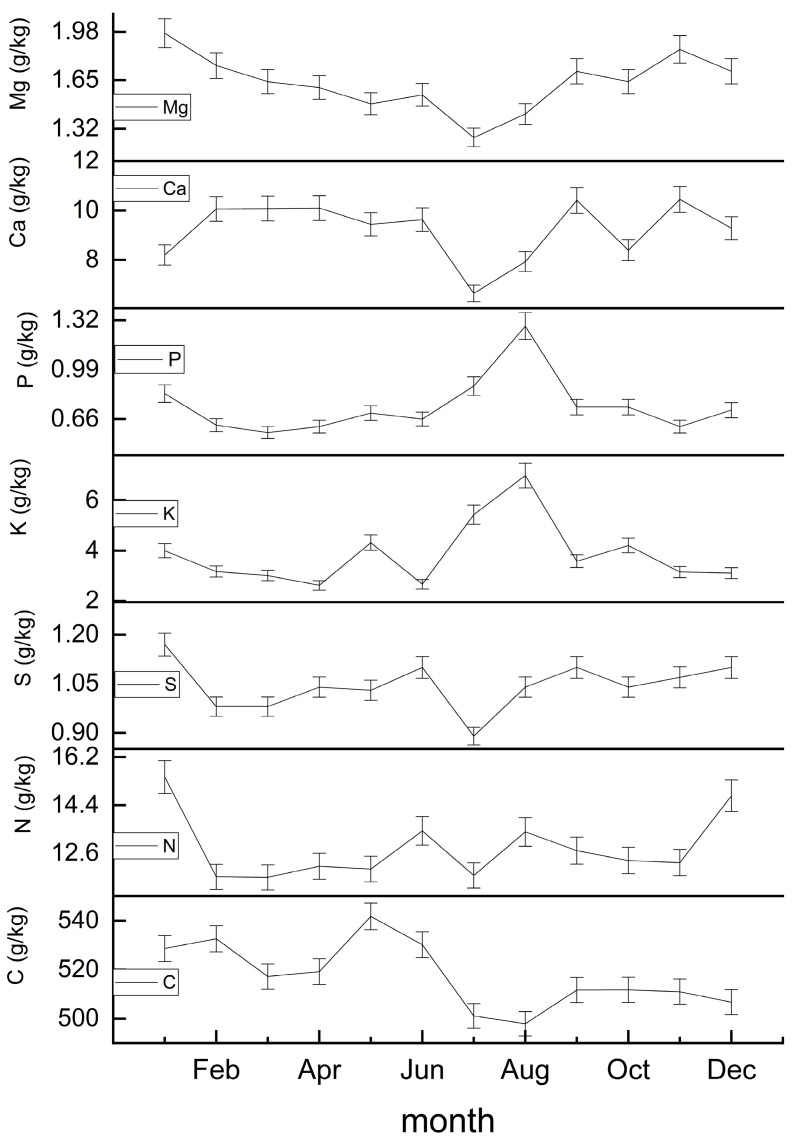
Within-year nutrient return of each element.

**Figure 5 plants-12-01277-f005:**
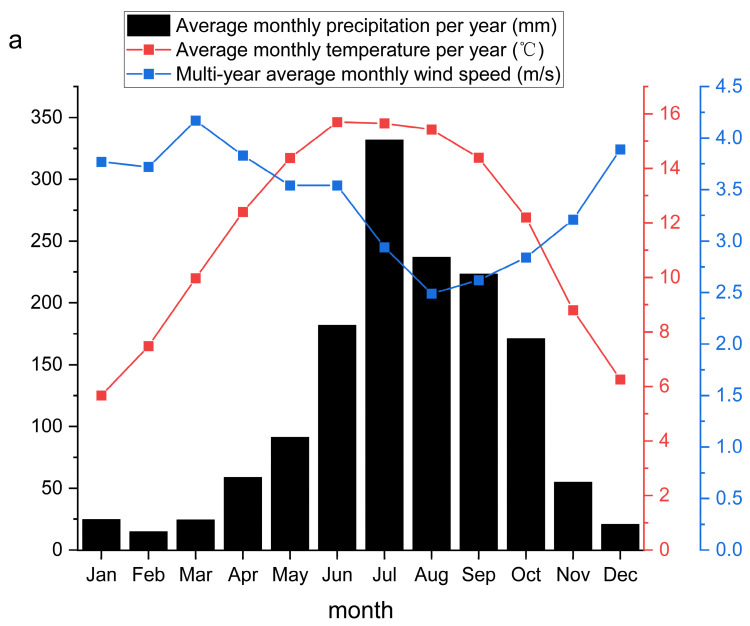
(**a**) Ailao Mountain Meteorological Station has average monthly precipitation, average monthly temperature for many years, average monthly wind speed for many years, (**b**) temperature, precipitation and wind speed from 2005 to 2017.

**Table 1 plants-12-01277-t001:** Annual total litterfall productions and components.

Year	Branches		Leaf		Others		Total
	(t/ha a)		(t/ha a)		(t/ha a)		(t/ha a)
2005	2.32 ± 0.80 Aa	26.9%	4.36 ± 0.52 Bb	52.0%	1.76 ± 0.12 Aa	21.1%	8.36 ± 0.88
2006	1.60 ± 0.20 Aa	19.5%	4.40 ± 0.84 Bb	54.6%	2.04 ± 0.24 Ba	25.9%	8.08 ± 1.16
2007	2.00 ± 0.60 Aa	22.9%	4.52 ± 0.64 Bb	52.8%	2.12 ± 0.16 Ba	24.3%	8.60 ± 1.00
2008	1.88 ± 0.36 Aa	23.2%	4.52 ± 0.8 Bb	52.5%	2.08 ± 0.28 Ba	24.3%	8.48 ± 1.12
2009	1.32 ± 0.20 Aa	19.7%	4.00 ± 0.48 Bb	59.1%	1.44 ± 0.12 Aa	21.2%	6.88 ± 0.60
2010	1.36 ± 0.24 Aa	17.6%	4.80 ± 0.84 Cb	61.5%	1.52 ± 0.20 Aa	20.9%	7.76 ± 1.20
2011	1.28 ± 0.16 Aa	18.6%	4.24 ± 0.56 Bb	59.5%	1.56 ± 0.08 Aa	21.8%	7.12 ± 0.52
2012	1.60 ± 0.20 Aa	18.9%	4.80 ± 0.84 Cc	55.6%	2.20 ± 0.28 Ca	25.5%	8.64 ± 1.04
2013	1.32 ± 0.20 Aa	17.3%	4.48 ± 0.60 Bc	57.5%	1.96 ± 0.28 Bb	25.2%	7.80 ± 0.80
2014	2.12 ± 0.52 Aa	22.5%	5.20 ± 1.12 Cb	54.1%	2.24 ± 0.28 Ca	23.3%	9.40 ± 1.44
2015	2.52 ± 1.40 Ba	32.0%	3.28 ± 0.44 Ab	41.7%	2.04 ± 0.32 Ca	26.3%	8.08 ± 1.92
Total average	1.76 ± 0.2	21.7%	4.42 ± 0.18	52.0%	1.91 ± 0.06	23.6%	8.11 ± 0.27

Note: Different uppercase letters indicate significant differences in litterfall output between different years of the same component (*p* < 0.05), and different lowercase letters indicate significant differences in litterfall output of different components in the same year (*p* < 0.05).

**Table 2 plants-12-01277-t002:** Intra-year variation of litterfall productions and components.

Month	Branches		Leaf		Others		Total
	(t/ha a)		(t/ha a)		(t/ha a)		(t/ha a)
January	0.33 ± 0.54 Aa	41.04%	0.26 ± 0.15 Aa	32.10%	0.22 ± 0.18 Aa	26.86%	0.82 ± 0.76
February	0.3 ± 0.63 Aa	41.17%	0.28 ± 0.18 Aa	37.74%	0.15 ± 0.11 Aa	21.09%	0.73 ± 0.8
March	0.2 ± 0.23 Aa	17.22%	0.77 ± 0.26 Bb	67.93%	0.17 ± 0.1 Aa	14.85%	1.14 ± 0.4
April	0.29 ± 0.53 Aa	15.62%	1.22 ± 0.35 Cb	66.49%	0.33 ± 0.15 Ca	17.89%	1.85 ± 0.75
May	0.12 ± 0.21 Aa	10.82%	0.78 ± 0.24 Bb	66.57%	0.27 ± 0.13 Ba	22.61%	1.16 ± 0.37
Jun	0.21 ± 0.57 Aa	26.68%	0.4 ± 0.18 Aa	47.65%	0.22 ± 0.19 Aa	25.67%	0.84 ± 0.81
July	0.16 ± 0.32 Aa	26.58%	0.28 ± 0.09 Aa	40.76%	0.22 ± 0.34 Aa	32.66%	0.66 ± 0.6
August	0.22 ± 1.13 Aa	31.69%	0.31 ± 0.17 Aa	42.13%	0.19 ± 0.25 Aa	26.18%	0.74 ± 1.45
September	0.2 ± 0.33 Aa	27.19%	0.36 ± 0.12 Aa	45.86%	0.21 ± 0.23 Aa	26.95%	0.79 ± 0.45
October	0.14 ± 0.29 Aa	19.67%	0.38 ± 0.12 Ab	49.55%	0.24 ± 0.25 Aa	30.78%	0.77 ± 0.43
November	0.1 ± 0.16 Aa	11.98%	0.55 ± 0.17 Aa	59.74%	0.26 ± 0.36 Ba	28.27%	0.92 ± 0.46
December	0.07 ± 0.07 Aa	10.62%	0.49 ± 0.2 Ab	66.43%	0.17 ± 0.19 Aa	22.96%	0.73 ± 0.3
Total average	0.19 ± 0.5	23.36%	0.51 ± 0.28	51.91%	0.22 ± 0.18	24.73%	0.93 ± 0.74

Note: Different uppercase letters indicate significant differences in litterfall output between different months of the same component (*p* < 0.05), and different lowercase letters indicate significant differences in litterfall output of different components in the same month (*p* < 0.05).

**Table 3 plants-12-01277-t003:** Correlation coefficients between litterfall productions of different components and various meteorological factors.

	Average Monthly Wind Speed	Monthly Precipitation	Monthly Precipitation Maximum	Average Monthly Temperature
Total	−0.127 *	0.166 **	0.158 **	0.139 *
Branches litter	−0.049	0.065	0.064	0.042
Leaf litter	−0.092	0.108	0.099	0.165 **
Others	−0.185 **	0.197 **	0.177 **	0.132 *

Note: * Significantly correlated at level 0.05 (two-tailed). ** Significant correlation at level 0.01 (double-tailed).

**Table 4 plants-12-01277-t004:** Correlation coefficients between meteorological factors from different months and monthly total litterfall productions.

Month	Average Monthly Wind Speed	Monthly Precipitation	Monthly Precipitation Maximum	Average Monthly Temperature
January	−0.565	0.996 **	0.963 **	−0.268
February	0.028	0.773 **	0.717 *	−0.424
March	0.125	−0.332	0.085	0.497
April	0.419	−0.183	−0.291	0.406
May	0.103	0.190	0.012	−0.430
June	−0.468	−0.038	−0.012	−0.529
July	−0.028	−0.396	−0.240	0.303
August	−0.337	0.375	−0.121	0.240
September	−0.143	0.473	0.436	0.231
October	−0.308	0.384	0.317	−0.173
November	−0.056	0.291	0.255	−0.005
December	0.023	−0.245	−0.151	−0.178

Note: * Significantly correlated at level 0.05 (two-tailed). ** Significant correlation at level 0.01 (double-tailed).

**Table 5 plants-12-01277-t005:** Characteristics of annual average nutrient concentration of litterfall in different years.

Nutrient Element	2005		2010		2015	
	Branches (g/kg)	Leaf (g/kg)	Branches (g/kg)	Leaf (g/kg)	Branches (g/kg)	Leaf (g/kg)
C	526.44 ± 2.94 Cd	546.25 ± 2.45 Dc	489.31 ± 3.28 Ad	507.25 ± 2.78 Bc	486.63 ± 2.52 Ad	500.7 ± 2.7 Bd
N	9.13 ± 0.22 Ab	13.23 ± 0.27 Cb	10.22 ± 0.56 Bb	13.81 ± 0.64 Cb	8.52 ± 0.49 Ab	14.7 ± 0.5 Dc
P	0.48 ± 0.01 Aa	0.72 ± 0.02 Ba	0.49 ± 0.02 Aa	0.74 ± 0.04 Ba	0.5 ± 0.03 Aa	0.84 ± 0.04 Ca
K	1.99 ± 0.11 Aa	4.95 ± 0.17 Ca	1.8 ± 0.12 Aa	4.55 ± 0.23 Ca	2.66 ± 0.37 Ba	5.32 ± 0.17 Da
S	0.87 ± 0.02 Ba	1.24 ± 0.02 Ca	0.92 ± 0.03 Ba	1.25 ± 0.02 Ca	0.74 ± 0.04 Aa	1.25 ± 0.04 Ca
Ca	15.19 ± 0.43 Dc	13.08 ± 0.2 Bb	14.38 ± 0.81 Cc	12.19 ± 0.28 Bb	11.78 ± 0.6 Bc	10.67 ± 0.53 Ab
Mg	1.55 ± 0.05 Aa	2.55 ± 0.03 Ba	1.54 ± 0.05 Aa	2.47 ± 0.09 Ba	1.45 ± 0.09 Aa	2.47 ± 0.06 Ba
Total average	555.65 ± 3.78	582.02 ± 3.16	518.66 ± 4.87	542.26 ± 4.08	512.28 ± 4.14	535.95 ± 4.04

Note: Different uppercase letters in the same row indicate significant differences between different years (*p* < 0.05), and different lowercase letters in the same column indicate significant differences between different nutrient elements in the same year (*p* < 0.05).

**Table 6 plants-12-01277-t006:** Characteristics of annual average nutrient return of litterfall in different years.

Nutrient Element	2005		2010		2015	
	Branches (g/kg)	Leaf (g/kg)	Branches (g/kg)	Leaf (g/kg)	Branches (g/kg)	Leaf (g/kg)
C	542.27 ± 25.34 Bc	553.96 ± 24.60 Bc	491.33 ± 9.67 Ad	509.16 ± 16.44 Ad	481.84 ± 14.29 Ac	500.20 ± 10.59 Ae
N	10.30 ± 2.07 Ab	13.72 ± 1.66 Bb	9.98 ± 2.42 Ac	17.60 ± 5.16 Dc	8.50 ± 2.13 Ab	15.81 ± 2.30 Cd
P	0.51 ± 0.15 Aa	0.66 ± 0.12 Ba	0.66 ± 0.40 Ba	1.02 ± 0.50 Ca	0.73 ± 0.33 Ba	0.98 ± 0.24 Ca
K	2.17 ± 1.14 Aa	3.73 ± 0.86 Ba	3.25 ± 2.90 Bb	4.95 ± 2.96 Cb	3.79 ± 2.49 Ba	5.13 ± 2.23 Cb
S	0.93 ± 0.21 Ba	1.18 ± 0.09 Ca	0.76 ± 0.17 Aa	1.27 ± 0.17 Ca	0.74 ± 0.19 Aa	1.25 ± 0.15 Ca
Ca	11.16 ± 2.32 Cb	9.29 ± 1.24 Bb	8.81 ± 3.84 Bc	7.53 ± 2.81 Ab	10.43 ± 4.39 Cb	7.98 ±1.99 Ac
Mg	1.37 ± 0.36 Ba	1.96 ± 0.31 Ca	1.07 ± 0.37 Aa	1.80 ± 0.40 Ca	1.29 ± 0.44 Ba	1.88 ± 0.37 Ca
Total average	568.71 ± 26.20	584.5 ± 24.48	515.86 ± 17.83	543.33 ± 18.50	507.32 ± 18.05	533.23 ± 11.47

Note: Different uppercase letters in the same row indicate significant differences between different years (*p* < 0.05), and different lowercase letters in the same column indicate significant differences between different nutrient elements in the same year (*p* < 0.05).

**Table 7 plants-12-01277-t007:** Correlation coefficient between meteorological factors and nutrient return.

Nutrient Element	Temperature		Precipitation		Wind Speed	
	*p*	r^2^	*p*	r^2^	*p*	r^2^
C	0.604	0.028	0.098	0.249	0.069	0.294
N	0.185	0.169	0.558	0.035	0.737	0.012
P	0.259	0.125	0.052	0.327	0.023 *	0.421
K	0.165	0.183	0.028 *	0.397	0.014 *	0.466
S	0.267	0.122	0.262	0.124	0.795	0.007
Ca	0.435	0.062	0.045 *	0.345	0.222	0.145
Mg	0.001 **	0.655	0.008 **	0.527	0.247	0.131

Note: * indicates significant correlation (*p* < 0.05), ** indicates very significant correlation (*p* < 0.01).

**Table 8 plants-12-01277-t008:** Biological cycle of different nutrients.

Nutrient Element	*R* _e_	*R* _g_	*T*_t_ (a)
C	0.25	0.50	9.93
N	0.24	0.52	9.32
P	0.23	0.84	8.40
K	0.24	0.53	9.09
S	0.25	0.49	10.30
Ca	0.29	0.42	14.14
Mg	0.28	0.44	12.80
average	0.25	0.53	10.50

## Data Availability

The data are not publicly available due to the dataset is proprietary and the author currently does not have the rights to make the data public.

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
