# Peer review of "Temporal Changes in Litterfall and Nutrient Cycling from 2005–2015 in an Evergreen Broad-Leaved Forest in the Ailao Mountains, China"

_plants, 2023, doi:10.3390/plants12061277_

Round 1

Reviewer 1 Report

The current study measured 11 years of litterfall in a wet evergreen broadleaved forest in Ailao Mountains southwestern China. Such a long period of measurements in monthly intervals requires researchers-energy, good budget, and lot of laboratory work. Long-term studies, such as this one, have much merit by analyzing the inter-annual variability of process such as litterfall and nutrient uptake by plants and nutrient return to the forest floor, in face of climate change. However, I think the current version of manuscript needs substantial improvement before publication, especially for the data analysis and presentation.

There are many part of the manuscript that need deep revisions. The authors need to revise the technical terms associated to this ecological process (litterfall, nutrient uptake, nutrient return in litterfall) for better using them in the manuscript. I would suggest the use of litterfall (as a ecological process) instead of “litter”, to avoid confusing with forest floor litter. Please note the difference between nutrient concentration and nutrient content and use the specific term in the manuscript. Homogenize the use of terms, such as wind speed or wind velocity. If possible, use hectare  (ha) instead hm2.

Line 111-113. Provide more information on the statistical analysis. What was the factor in the ANOVA?

Line 117. “.. Cij is the nutrient content (g/kg)”. Actually this is nutrient concentration.

How did you analyze the effect of environmental variables on litterfall?

I suggest to revise the Discussion section and discuss in terms of ecological implications of the results, rather than only compare the results with those reported in other studies.

Specific comments:

Line 85. Did you consider all branch size that fell into the litter traps? Or, did you sorted then until certain size (diameter)? For larch branches, 1m2 litter traps could be too small for good precision in biomass estimation.

Line 124. What is “nutrient regression?

Line 263-264. “The nutrient content of litter depends on the uptake of soil nutrients by vegetation, and the uptake of nutrients by vegetation” This sentence clearly needs revision.

Line 279-280. Homogenize the use of units, tones or kilograms.

Line 283-285. Avoid speculation on your results. Stating that the studied forest “has the potential of strong carbon sink” is not supported by your results.

Line 322-326. Write the conclusion based on your objectives avoiding speculations.  

Fig. 1. I suggest to name the two graphs with A and B to better explain in the graphs declarative title. How many years are included in “… for many years”? .

Fig. 2. Change Litter Drop in Y axis and in the legend by litterfall.

Fig. 3. Each panel named y letters must have an explanation in the Fig declarative title.

Table 5. What are the units of nutrient content?

Fig. 4. What are the units in axis Y for each panel?

Table 6. What are the units of nutrient content?

Table 7. What is “T”? R should be r (correlation coefficient)

Table 8. What are the units of nutrient content?

Author Response

Dear Reviewer:

Thank you for your valuable comments and suggestions on my manuscript, I have revised the manuscript according to your suggestions, and answer the questions you raised below.

I have modified the usage of units and specific terms in the manuscript, and have repeatedly confirmed and ensured its standardization, including litterfall, concentration, hectare, etc. Detailed descriptions of statistical analysis related information are also included in the manuscript. We averaged the total litterfall output across the 25 collection baskets, and took monthly and annual (sum of the 12 months) for analyses. After using the Shapiro–Wilk test to test the normality of the data, one-way ANOVA and LSD were used to compare the difference in the amount of litterfall in different parts of different years and its components, the concentration of nutrient elements, and the amount of return. The ANOVA of the article is mainly used to test whether there is a significant difference in the mean of the components of litterfall between the Interannual and Intra-annual periods, and present the P value in the chart below. The use of multiple comparisons can better understand the differences between them; in addition, we also use SPSS26.0 to linear fit the environmental variables (temperature, precipitation, wind speed) and the output of components of litterfall and nutrient concentration, and explore the correlation between them.

You mentioned this question in your reply, “How did you analyze the effect of environmental variables on litterfall?” I use statistical methods to analyze the effect of environmental variables on litterfall. First, I collected data on environmental variables, including temperature, precipitation and wind speed; then I used regression analysis to explore the relationship and correlation between environmental variables and litterfall, and finally tested whether the analysis results were statistically significant. In addition, I have also made detailed revisions to the Introduction, discussion and conclusion sections of my manuscript in accordance with your suggestions, which are also presented in the new manuscript.

You also mentioned such a question, “Did you consider all branch size that fell into the litter traps? Or, did you sorted then until certain size (diameter)? For larch branches, 1m2 litter traps could be too small for good precision in biomass estimation.” My answer is that we did not consider the size of the branches, nor did we sort the branches by size. What we mainly considered was the dry weight of each component of the litterfall and the amount of nutrients it contained. But your mention of this detail makes me think more about the design of the experiment. For litter traps, we set up 25 litter traps in a 1-hectare plot. We carefully considered the location of the 25 litter traps to make them represent the actual situation of the entire plot as much as possible. Of course, compared with the actual data of the entire plot, there may be some errors, but overall within a controllable range. I apologize for the mistake of “nutrient regression”, what I meant to expression was “nutrien treturn”, and I have already revised the content in the manuscript. In my communication with my supervisor, I found T-value is an unnecessary data and have deleted it in my new manuscript. T-value is the result of T-test, which is a statistical test used to test whether the means of two independent samples are significantly different. I spent some time to carefully revise the chart to make it clearly and accurately convey our research results, including units, professional terms, legends, resolutions, etc.

Finally, thank you again for your valuable suggestions on my manuscript, which makes my manuscript more perfect. Wish you a smooth work.

                                                                                                                           Dai

Reviewer 2 Report

Dear Editor, dear Authors

the manuscript could be interesting for the scientific community and would reflect the aims of the journal; the Authors report data on litter fall and its nutrient composition during 11 years with data collected monthly. The duration of the study, the frequency of samplings and the number of measurements make the manuscript interesting, but the scientific novelty and importance of the results obtained are unclear. The Authors do not consider the importance of the species composition of forest. This is the strong limitation of the research presented in this paper because it is the monitoring of a specific forest.

In addition, the paper is not well written, the introduction is too general and does not actually report current knowledge. The results and discussion are very confusing. The figures are poorly presented and not all of them are necessary. Some figures and tables are not referred to in the text (Figure1, Table 2 and Table 4). The captions should be more detailed; units are missing in Figure 4. The Authors should also review “Abstract” and “keywords”. In the abstract the Authors should avoid a bulleted list; Some keywords are not relevant. The description of the statistical analyses used is not clear and is incomplete, e.g. the authors do not report correlations. The Authors report “litter fall” but also “litter drop”. Are there differences? Similarly, why to report “Yield”? What do the Authors mean by regressors?

The Authors should also correctly report their affiliations.

In conclusion, I believe the manuscript needs deep revisions before being considered for a new submission.

The manuscript should be rejected in the present version.

Author Response

Dear Reviewer:

Thank you for your valuable comments and suggestions on my manuscript, I have revised the manuscript according to your suggestions, and answer the questions you raised below.

Our manuscript presents our results, which show that litterfall plays a very important role in the entire forest ecosystem, providing a channel for nutrient cycling. At the same time, it plays a fundamental role in maintaining soil fertility and biodiversity in the study area. I believe that these data provide a fundamental data support for further research in the study area, which is valuable content for the study area. For the question of species composition, our sample site is a well-preserved primary forest, where tall evergreen broad-leaved trees, herbaceous plants and moss-lichen shrubs are mostly present, and the litterfall collected from this site is mainly produced by these tall trees; special sample sites are set up for the collection of litterfall of specific species, including moss-lichen shrubs, herbaceous plants, etc.; in this paper, we only used the data from one sample site, and the litterfall collected only covered tree species, and we did not consider the differences between tree species when classifying the litterfall. The differences are not very obvious because the fallen materials collected mainly include Machilusbombycina, Populusrotundifolia, Schimanoronhae, Castanopsisrufescens and Castanopsisdelavayi.

I have modified the usage of units and specific terms in the manuscript, and have repeatedly confirmed and ensured its standardization, including litterfall, concentration, hectare, etc. Detailed descriptions of statistical analysis related information are also included in the manuscript. We averaged the total litterfall output across the 25 collection baskets, and took monthly and annual (sum of the 12 months) for analyses. After using the Shapiro–Wilk test to test the normality of the data, one-way ANOVA and LSD were used to compare the difference in the amount of litterfall in different parts of different years and its components, the concentration of nutrient elements, and the amount of return. The ANOVA of the article is mainly used to test whether there is a significant difference in the mean of the components of litterfall between the Interannual and Intra-annual periods, and present the P value in the chart below. The use of multiple comparisons can better understand the differences between them; in addition, we also use SPSS26.0 to linear fit the environmental variables (temperature, precipitation, wind speed) and the output of components of litterfall and nutrient concentration, and explore the correlation between them. In addition, I have also made detailed revisions to the Introduction, discussion and conclusion sections of my manuscript in accordance with your suggestions, which are also presented in the new manuscript.

I apologize for the mistake of “nutrient regression”, what I meant to expression was “nutrien treturn”, and I have already revised the content in the manuscript.I spent some time to carefully revise the chart to make it clearly and accurately convey our research results, including units, professional terms, legends, resolutions, etc.

Finally, thank you again for your valuable suggestions on my manuscript, which makes my manuscript more perfect. Wish you a smooth work.

                                                                                                                            Dai

Round 2

Reviewer 1 Report

The manuscript is much in better shape now. Most of my comments were addressed. However, there are still some inconsistencies that need revision.

Figures and Tables need more edits. Labels seems to be too small.

Edit equation 1. Why the sum is divided by 100?

Equations 2-4 need more explanation. How the terms in each equations are derived? Unfortunately the cite (Zhang et al, 2019) seems to be in Chinese. I could not get the paper to review the equations.

Line 148: “La is the amount of annual litterfall returned” What does this mean? Litterfall production? Or, is it the amount of certain nutrient?

Table 1. “production” should be instead of “productions”. Remove litterfall before branches and leaves. Remove output and leave only (t.ha-1 a-1).

The correct notation is kg ha-1 or kg ha-1 a-1.  Please homogenize throughout  the manuscript. For example in Figure 3, the Y axis title is “t/ha.a-1”. If you want to use this notation, the correct form is t/ha a. But I insist, the same notation should be used wether in Figures, Tables, or text.

What is the difference between Table 5 and Table 6? The values area similar, and so area the units.

Figure 5. Within-year regressors of each element. What does this mean?

Author Response

Dear Reviewer:

Thank you for your valuable comments and suggestions on my manuscript, I have revised the manuscript according to your suggestions, and answer the questions you raised below.

I have made more detailed revisions to the graphs and tables according to your suggestions, and also adjusted the labels to a size that is easy for readers to read. In addition, detailed revisions have been made to all units in the manuscript according to your suggestions, which involve graphics, tables, and texts. My explanation for the issue with Formula (1) is that 100 is not the sum divided by 100, it is used to represent the percentage with Cij. When I wrote the formula, I thought it had no effect outside the brackets, but it contradicted what I intended to convey. Therefore, I made some adjustments to it (removing the brackets). I apologize for any misunderstanding caused by my lack of accuracy in expressing what I wanted to say about La, which is the annual amount of nutrients returned by Litterfall. Formula2-4, in my latest manuscript, I have introduced the explanation of the equations in a relatively detailed way, as you said, the detailed derivation process should be expressed, which is also conducive to the reader's reading.

My interpretation for Tables 5 and 6 is that 5 contains the nutrient concentration of litterfall, while 6 contains the nutrient return of litterfall. Although table 5 and table 6 have the same presentation format, they express different meanings. I believe that presenting more data with two tables makes it easier for readers to understand. “What about the within-year regressors in Figure 5?” Regarding this issue, I am very sorry that this issue was not handled well in the last round of revisions, and I want to express the “nutrient return”.

Finally, thank you again for your valuable suggestions on my manuscript, which makes my manuscript more perfect. Wish you a smooth work.

Reviewer 2 Report

Dear Editor, dear Authors,

The manuscript improved in the revision and certainly the most sensitive points were revised and clarified by the authors. Despite this I still have a few suggestions for the authors. The keywords continue to be repetitive and are included in the title.

The changes in the introductory part are more convincing than those in the discussion of the data. I advise Authors to make comparisons with other data collected even in different systems.

Authors should follow the journal's instructions for citing references in the text and listing them in the manuscript. The figures need to be changed and the captions are still lacking in details. 

Authors should follow the journal's instructions for citing references in the text and listing them in the manuscript. The figures need to be changed and the captions are still lacking in details.

In Figures 2 and 3, the parameters on the y-axis must be entered, before the units in brackets. In Figure 2, the Authors should delete the legend. In the manuscript, the Authors have replaced the word 'litter' with 'litterfall', but this is not always correct. For instance, when the Authors mention the different components of litter, it would be more appropriate to use 'leaf litter', 'branches litter' and 'others'. This correction should concern both the text and the tables.

In conclusion, I believe the manuscript needs minor revisions before being considered for the submission.

Author Response

Dear Reviewer:

Thank you for your valuable comments and suggestions on my manuscript, I have revised the manuscript according to your suggestions, and answer the questions you raised below.

I have reorganized my content and made detailed modifications to the keywords so that it can echo the title and achieve comprehensive expression of my manuscript content. I also made more detailed changes to the title according to your suggestion in order to express the content of my manuscript more detailedly. I am thankful to you for the valuable suggestions you have given me regarding the discussion part of the manuscript. I have already revised it according to your suggestions. The revision also involved more detailed editing of graphics, tables, units, words, etc. For example, deleting the legend of Figure 2, the Y-axis parameters and label sizes of Figure 3, etc.

Finally, thank you again for your valuable suggestions on my manuscript, which makes my manuscript more perfect. Wish you a smooth work.